# Novel Pharmacological Treatment Options in Pediatric Glioblastoma—A Systematic Review

**DOI:** 10.3390/cancers14112814

**Published:** 2022-06-06

**Authors:** Johanna Wyss, Nicole Alexandra Frank, Jehuda Soleman, Katrin Scheinemann

**Affiliations:** 1Division of Oncology-Hematology, Department of Pediatrics, Kantonsspital Aarau, 5001 Aarau, Switzerland; katrin.scheinemann@ksa.ch; 2Division of Pediatric Oncology-Hematology, University Children’s Hospital of Basel, 4056 Basel, Switzerland; 3Department of Neurosurgery, University Hospital of Basel, 4031 Basel, Switzerland; nicolealexandra.frank@usb.ch (N.A.F.); jehuda.soleman@usb.ch (J.S.); 4Department of Pediatric Neurosurgery, University Children’s Hospital of Basel, 4056 Basel, Switzerland; 5Faculty of Medicine, University of Basel, 4056 Basel, Switzerland; 6Department of Health Sciences and Medicine, University of Lucerne, 6002 Lucerne, Switzerland; 7Department of Pediatrics, McMaster University Hamilton, Hamilton, ON L8S 4K1, Canada

**Keywords:** pediatric glioblastoma, novel therapeutics, treatment resistance, targeted therapy

## Abstract

**Simple Summary:**

Childhood glioblastoma is an aggressive brain tumor in children that has a very poor prognosis. Standard therapy includes surgery, irradiation and chemotherapy with temozolomide. So far, there is no effective drug treatment for pediatric glioblastoma patients. This systematic review aims to outline currently available data on novel pharmacological treatment options. None of the included phase II studies showed any benefit regarding overall survival or a prolongation of stable disease. New genomic technologies discovered the biologic heterogeneity of these tumors, demanding more individualized immunotherapeutic and targeted approaches. Autoimmune modulated therapies and further targeting of tumor-specific receptors provide promising preclinical results. Clinical trials aligned to the tumor characteristics are needed to establish effective new therapeutic approaches.

**Abstract:**

Background: Pediatric glioblastoma (GBM) is an aggressive central nervous system tumor in children that has dismal prognosis. Standard of care is surgery with subsequent irradiation and temozolomide. We aimed to outline currently available data on novel pharmacological treatments for pediatric GBM. Methods: We conducted a systematic literature search in PubMed and Embase, including reports published in English from 2010 to 2021. We included randomized trials, cohort studies and case series. Phase I trials were not analyzed. We followed PRISMA guidelines, assessed the quality of the eligible reports using the Newcastle-Ottawa scale (NOS) and the RoB-2 tool and registered the protocol on PROSPERO. Results: We included 6 out of 1122 screened reports. All six selected reports were prospective, multicenter phase II trials (five single-arm and one randomized controlled trial). None of the investigated novel treatment modalities showed any benefit regarding overall or progression free survival. Conclusions: To date, the role of pharmacological approaches regarding pediatric GBM remains unclear, since no novel treatment approach could provide a significant impact on overall or progression free survival. Further research should aim to combine different treatment strategies in large international multicenter trials with central comprehensive diagnostics regarding subgrouping. These novel treatment approaches should include targeted and immunotherapeutic treatments, potentially leading to a more successful outcome.

## 1. Introduction

Central nervous system (CNS) tumors are the most common form of solid tumors in children and account for the majority of cancer mortality in this age group [1]. Pediatric CNS tumors represent a heterogenous group with different histology, molecular varieties and biological behavior.

Pediatric glioblastoma (GBM) is most often localized supratentorial, whereas cerebral hemispheres account for approximately half of the cases [2]. Tumor location in the infratentorial region is associated with poor survival [3]. As an important first diagnostic tool, contrast enhanced magnetic resonance imaging (MRI) exhibits either rim or heterogeneous enhancement (Figure 1). Tumors with rim enhancement were shown to have better prognosis [4]. T1-native and contrast enhanced MRI provides information of a necrotic center of the tumor mass and areas with disrupted blood–brain barrier (BBB). T2-weight imaging provides information about the disease progress to peripheral structures due to higher water content in this tissue [5]. Diffusion-weight imaging (DWI) helps to distinguish between cerebral abscess and tumor-suspect lesions [6]. Further options to better classify the lesion, especially in deep located tumors that are difficult to biopsy, are magnetic resonance spectroscopy (MRS) since brain tumors demonstrate a reduced N-acetylaspartate (NAA) and creatine level and increased choline levels [7].

GBM is classified by the WHO as grade IV glioma, amplifying the aggressiveness and resistance to currently available treatment options [8]. Pediatric GBM accounts for 2.9% of all histologies amongst pediatric CNS tumors in the USA and is most prevalent in children from 10–14 years of age [1].

The poor prognosis of pediatric GBM is reflected by a median survival of only 13–43 months after diagnosis [9,10,11]. The five-year overall survival (OS) for children and adolescents diagnosed with GBM is <20% [12]. Different molecular, and hence outcome behaviors, can be observed in infant and congenital GBM. The definition of infant GBM typically refers to children younger than three to five years of age [13], whereas congenital GBM is defined as presence at birth and represents the most seldom type [14]. Congenital GBM shows the worst prognosis limited to approximately two months due to higher tendency of intracranial hemorrhage [15], while infant GBM tends to differ in clinical outcome from pediatric patients even with incomplete resection with improved five-year OS of 66% [16,17]. However, early relapse in infant GBM is frequent, reflected by a five-year event free survival (EFS) of less than 30% [16,18].

The standard of care for pediatric GBM older than three years of age is gross total resection (GTR) with subsequent irradiation, typically with 50–60 Gy, and temozolomide (TMZ), currently offering the best OS [11,19,20]. At times GTR is not feasible, subtotal resection (STR) is offered, since it was shown to improve OS and progression free survival (PFS) as well [21,22]. TMZ, an inherent part of GBM treatment in adults in combination with irradiation [23], showed lower toxicity than other drug regimens with comparable effectiveness and was therefore added to the standard treatment [24]; although improved outcome could not be shown with the addition of TMZ [25,26]. TMZ is an alkylating agent, which works most effectively in patients with methylated *MGMT* promoter. However, it was shown that pediatric GBM tumor cells display *MGMT* promoter methylation significantly less often [26,27] and, therefore, showed a less effective response to TMZ as expected [25,26]. Children younger than three years of age receive surgery as standard treatment and chemotherapy if feasible [28]. Irradiation is not recommended due to severe neurocognitive sequelae and is often not mandatory initially, due to a better response to chemotherapy than older children [16]. Most patients still require irradiation in the relapse situation and long-term sequelae are severe, as these children are often still very young [16].

Pediatric GBM have a high molecular heterogeneity compared to adult GBM but also within its group. Six distinct epigenetic and biological subgroups of pediatric and adult GBM have been defined through DNA methylation studies [29,30]. These six methylation clusters include K27, G34, IDH, receptor tyrosine kinase (RTK) I and II and mesenchymal cluster. They show a distinct age distribution. The K27 (Lys27Met) cluster appeared predominantly in the pediatric population, while the G34 (Gly34Arg) cluster was more frequently detected in adolescents [1]. The most common somatic alterations in infant high-grade glioma (HGG) involve neurotrophin receptor tyrosine kinase 1/2/3 (*NTRK1/2/3*) genes and have been described in 40% of non-brainstem HGG in infants, including infant GBM [2]. The other clusters were less specific for the pediatric age group [1] (Table 1). Hence, different possible targets have been the focus of research for new therapeutic approaches during the past decade. The K27 and G34 clusters are *H3*-mutant gliomas. *H3*K27M induces histone modifications that can be targeted directly by histone deacetylase (HDAC) or demethylation inhibitors [31,32,33]. *H3*G34R/V gliomas result in an upregulation of MYCN. PDGFRA mutations were identified as major drivers in *H3*G34R/V mutant gliomas, leading to a further therapeutic target [34]. Synergy was demonstrated for the PDGFRA antagonist dasatinib in combination with the mTOR inhibitor everolimus in pediatric patients [35]. *H3*- and *isocitrate dehydrogenase* (IDH) wildtype gliomas show distinct molecular subtypes with mutations such as *BRAF*V600E or RTK fusions (such as NTRK fusions). *BRAF*- and mitogen-activated protein kinase kinase (MEK) inhibitors showed promising results in pediatric LGG [36,37,38] while efficacy in HGG is less well understood. Tropomyosin receptor kinase (TRK) fusion-positive gliomas can be targeted with larotrectinib, a highly selective small-molecule inhibitor of TRK fusion positive gliomas [39]. A further subgroup is EGFR amplificated glioma, where newer EGFR inhibitors showed improved BBB penetration with better clinical efficacy in salvage therapy [40,41]. Further novel approaches to pediatric GBM are immunotherapeutic strategies, such as cancer vaccines, monoclonal antibodies, immune checkpoint inhibitors and chimeric antigen receptor (CAR)-T cells.

The aim of this systematic review is to outline currently available data on novel pharmacological treatment options for pediatric GBM.

## 2. Materials and Methods

For this systematic review, we searched PubMed and Embase databases and included reports published in English between January 2010 until December 2021. We defined 2010 as the start date of the assessed reports, since while screening the years before, a substantial lack of new treatment modalities next to TMZ was seen, which was implemented as standard treatment for adult GBM in 2005. Our utilized search string included the search items “glioblastoma” and “pediatric” and “drug treatment” and “chemotherapy” (Figure 2a).

We included randomized trials, retrospective and prospective cohort studies, and case series including more than five pediatric patients with the diagnosis of GBM. Pediatric age was defined between 0 and 18 years of age. Technical reports, case reports, comments, editorial letters, poster abstracts and reviews were excluded from this report. Phase I trials focusing on pharmacokinetics and estimation of dose tolerance and limitation were excluded, as our intention was to focus on available data on effective treatment options for clinical use in pediatric GBM. Reports including only standard treatment with TMZ, and reports describing HGG without a differentiation into subgroups or without specific baseline characteristics regarding GBM were excluded as well.

After removal of duplicates, which was conducted with the help of the web-based software Rayyan [46], the results were screened by title by two authors, independently (J.W. and N.A.F.). Further, the abstracts were assessed followed by a full text evaluation of the remaining reports. In case of disagreement concerning the in- or exclusion of a report, the senior authors (J.S. and K.S.) made the final decision. One further report was included by screening the references of the other reports. We defined relevant parameters of the studies in reference to PICO standards (Appendix A). We extracted the following information from eligible reports: study details (author, year of publication, design and statistics); study population (number of participants, recruitment interval, median age); treatment characteristics (disease status, intervention); outcome measures (EFS, OS, PFS, toxicity and toleration). The primary outcome measure was overall outcome with EFS, PFS or OS. Secondary outcome measures were toxicity and toleration.

The included reports were classified in “newly diagnosed” and “recurrent or refractory” GBM. “Recurrent” is defined by GBM relapse after first- or second-line treatment with initial objective response, whereas “refractory” is defined by progressive GBM failing to respond to treatment. PFS is defined by the length of time during and after treatment without radiological or clinical progression. EFS is defined as the interval between treatment and documentation of clinical or radiological disease progression, complications from disease or treatment, secondary malignancy or death of any cause.

This systematic review was performed in accordance with the Preferred Reporting Items for Systematic Reviews and Meta-Analyses (PRISMA) guidelines (Appendix A) and is registered in PROSPERO (Prospero-ID: CRD4202232200) (Figure 2b). Quality assessment of the studies was carried out by two reviewers (J.W. and N.A.F.) independently using the Newcastle-Ottawa scale (NOS) [47]. The assessment for the included randomized controlled trial (RCT) [48] was carried out using the RoB-2 tool [49].

## 3. Results

The systematic search of the databases yielded 1121 reports, while one report was added through screening of references. After removal of duplicates, 1019 report titles were screened and 53 reports assessed for abstract screening. Of these, 23 reports underwent full text evaluation whereof six reports were included in the qualitative analysis [48,50,51,52,53,54] (Figure 2b; Table 2). Four of the included reports investigated novel pharmacologic treatment strategies in therapy refractory or recurrent GBM patients [50,51,52,54], and two in newly diagnosed GBM patients [48,53]. All six selected reports were prospective, multicenter phase Ⅱ trials, five of them single-arm trials [50,51,52,53,54] and one of them a RCT [48]. In total, we included 137 pediatric patients receiving pharmacological treatment for pediatric GBM.

The investigated medications of the six included prospective studies were bevacizumab (BEV) as monotherapy and in combination with irinotecan [48,51], valproic acid (VPA) [53], cilengitide [50] and an oral combination therapy of thalidomide, celecoxib, fenofibrate, low dose etoposide and cyclophosphamide [52] and sunitinib [54].

The qualitative assessment for the five cohort studies showed a mean Newcastle-Ottawa scale (NOS) rating of 5 ± 0.4 (Table 2). The included RCT [48] was assessed as high quality with some concerns using the RoB-2 tool.

### 3.1. Newly Diagnosed Pediatric Glioblastoma

Grill et al. concluded a phase II, open-label, randomized, international comparator trial (HERBY trial) with the intervention of the addition of BEV to irradiation and TMZ in pediatric patients with newly diagnosed HGG [48]. Eighty-four (69%) of 121 included HGG patients were diagnosed with a pediatric GBM. EFS for the whole cohort in the BEV plus RT + TMZ was 11.8 months (95% CI 7.8–12.7 months) compared to 8.2 months in the cohort without BEV, showing no benefit for BEV in combination with irradiation and TMZ for GBM [48]. No detailed outcome information for the GBM cohort was made. MGMT methylation status was balanced between the intervention groups [48].

Meng-Fen Su et al. conducted a multi-institutional, single-arm phase Ⅱ clinical trial of irradiation and VPA, followed by maintenance with VPA and BEV in children with newly diagnosed HGG [53]. Out of 38 HGG patients enrolled for the study, 11 (28%) GBM patients were assessed showing a median EFS of 10.5 months and a median OS of 14.9 months. The estimated one-year EFS for all the HGG patients was 24%.

In conclusion, the addition of VPA and BEV to irradiation could not show a significant benefit for newly diagnosed pediatric GBM.

### 3.2. Recurrent or Refractory Pediatric Glioblastoma

In a phase Ⅱ study, Gururangan, et al. investigated the efficacy of the combination of BEV and irinotecan in children with recurrent HGG [51]. They evaluated 31 patients of which 8 patients (26%) were histologically diagnosed with a pediatric GBM. The primary objective of the study was to determine the objective response (complete response plus partial response) to BEV and irinotecan in recurrent pediatric HGG. Of the whole study cohort, no sustained objective responses were observed. Eight of these 31 patients (25%) showed a sustained stable disease at >12 weeks, while 23 (74%) patients had progressive disease. The group of the sustained stable disease patients included two GBM patients. These two patients showed a median PFS of 8.3 months with a median PFS of the whole glioma-cohort of 4.2 months.

In a phase Ⅱ study, MacDonald, et al. investigated the efficacy of cilengitide in the treatment of recurrent or refractory pediatric HGG [50]. Thirty patients were enrolled. Twenty-four patients were accessible for primary outcome analysis (six excluded due to severe progression prior to first MRI), of which 18 (75%) were diagnosed as GBM. The primary objective was to determine the objective response rate to cilengitide, defined as successful by complete (CR) or partial response (PR) or PFS for at least 12 weeks. Only one patient (4%) of the evaluable 24 patients showed a response with stable disease at 280 days. This response was in a young GBM patient < 3 years of age. For the remaining 23 patients, median time to progression was 28 days. Mortality occurred in 21 patients (87%) with a median time to death of 172 days [50].

Robinson et al. investigated in an open-label, single-arm, multi-institutional phase Ⅱ study the efficacy of an antiangiogenic, metronomic oral drug regimen with thalidomide, celecoxib, fenofibrate and low dose etoposide and cyclophosphamide in recurrent or refractory pediatric cancer [52]. They enrolled 101 pediatric cancer patients, of which 97 commenced treatments. All types of pediatric cancers were included; therefore, patients were categorized in seven strata. GBM patients were included in the HGG strata. Of these, nine were diagnosed with a pediatric GBM, with eight primary GBM and one secondary GBM.

The primary endpoint of the trial was the assessment of the effect of the five-drug regimen given over a period of 27 weeks. Secondary endpoints included OS and PFS. Of the 21 subjects in the HGG strata, 13 patients (62%) showed progressive disease, 7 (33%) stable disease and 1 patient (4%) showed partial remission. Eight out of eight (100%) of primary GBM patients were progressive, one patient showed a stable disease. This patient with a stable disease was diagnosed with a secondary GBM with a prior history of medulloblastoma.

In a multicenter phase Ⅱ trial, Wetmore, et al. evaluated the effect of sunitinib in the treatment of recurrent or refractory pediatric HGG and ependymoma [54]. Thirty patients were enrolled. Seventeen of them were diagnosed with a HGG, while seven of these were diagnosed with GBM. The primary objective of this study was to estimate overall response rate (ORR), defined as complete or partial response for at least eight weeks. The study had to be closed at the time of interim analysis due to missing sustained objective response. The observed response rate in the HGG cohort was 0% (95% Blyth-Still-Casella CI, 0−19.8%). Of the whole cohort of HGG, the median time to progression was 72 days (95% CI 33–84) [54].

In summary, no treatment modality could demonstrate a positive effect on OS or PFS in recurrent or progressive and pretreated pediatric GBM.

## 4. Discussion

Based on our systematic review of the literature, novel pharmacological treatment options for pediatric GBM showed no benefit in PFS or OS in the setting of newly diagnosed pediatric GBM as well as in recurrent or refractory GBM. Overall, there are only a few phase Ⅱ trials investigating novel drug treatments for pediatric HGG patients with a very limited number of pediatric GBM patients and short follow-up intervals, mainly due to the aggressive course of the disease.

### 4.1. Newly Diagnosed Pediatric Glioblastoma

We detected two reports investigating BEV and VPA in newly diagnosed pediatric GBM [48,53]. Both of them did not include infant GBM, as they both used concurrent irradiation. The lack of improvement of the outcome with BEV in children is not consistent with data from adult trials showing prolonged PFS [55]. BEV is approved in several countries worldwide for the treatment of relapsed GBM in adults [56]. This finding supports the emerging knowledge of molecular understanding of two different entities of pediatric and adult GBM tumor biologies and makes a translation of efficient treatment strategies from adult patients to pediatric patients impossible [57,58]. An important observation is the pattern of tumor recurrence after BEV treatment. A higher number of patients treated with the addition of BEV also showed, besides local recurrence, distant progression patterns [48,53,59]. The addition of the histone deacetylases (HDAC) inhibitor VPA and BEV to irradiation [53] could not improve the outcome either, although VPA showed a radio sensitizing effect in HGG patients [60,61] and promising results of partial response in a previous phase Ⅰ study in children [32]. The median EFS and OS of Su, et al. is most likely somewhat overestimated, as 36% of the GBM cohort were diagnosed with a mismatch repair deficiency (MMRD) syndrome, clearly exceeding the median EFS with the longest median EFS of 28.5 months [53]. One patient with a Lynch syndrome even showed a sustained complete remission, suggesting further treatment strategies with a HDAC or an alternative angiogenesis inhibitor should be investigated for this subgroup of pediatric GBM patients. MMRD GBM shows a differing genetic profile characterized by a high mutational burden, compared to conventional GBM, which results in different behavior regarding treatment [62]. Immune checkpoint inhibition is a further approach for this small subgroup showing promising results [62].

### 4.2. Recurrent or Refractory Pediatric Glioblastoma

For the relapsed or refractory setting of pediatric GBM, four reports were identified. The combination treatment of BEV and irinotecan did not show any sustained objective response [51]. These results are in contrast to the observed efficacy of the combination of BEV and irinotecan in adults, similar to the differences already mentioned with the use of single treatment with BEV [63,64,65]. The causes of treatment failure are multifactorial. Possible contributors could be alternative angiogenic pathways and resistance mechanisms maintaining tumor growth [66]. As a single agent, cilengitide showed no efficacy in pediatric GBM patients [50], in contrast to the previous phase I trial [67]. Cilengitide is an alpha(v) integrin antagonist demonstrated to block angiogenesis and showed a tumor regression of GBM cells in vivo [68,69,70]. The efficacy of cilengitide was shown in adult trials [71]. The results of adult trials with a combination of cilengitide, TMZ and irradiation showed significantly better results, especially in patients with methylated *MGMT* promotor status [71,72]. A synergistic effect of cilengitide and irradiation was also shown [73]. The very limited effect in pediatric patients might be due to the fact that pediatric GBM tumor cells display *MGMT* promotor methylation significantly less often [26,27]. Another treatment strategy are metronomic low dose treatment schedules of antiangiogenic and cytotoxic agents by suppressing endothelial cell proliferation and affecting the tumor microenvironment by rebuilding an anticancer immune response [74,75,76,77]. The primary endpoint is rather non-progression of disease than an objective tumor reduction and therefore presents another conception of therapy to treat cancer as a chronic disease with maintenance of quality of life [78]. However, the oral combination therapy of thalidomide, celecoxib, low dose etoposide/cyclophosphamide and fenofibrate, a PPAR-alpha agonist, showed an unfavorable response rate [52]. The only GBM patient with stable disease was diagnosed with a secondary GBM and can, therefore, biologically not be compared to primary GBM. The orally bioavailable sunitinib, a tyrosine kinase inhibitor (TKI) showed neither an objective response in pediatric patients [54] nor in adult recurrent GBM [79]. Sunitinib is an inhibitor of PDGFRα-β, vascular endothelial growth factor receptor (VEGFR1-2), fetal liver tyrosine kinase receptor 3 (FLT3) and stem cell factor receptor (KIT) [80]. Several tyrosine kinases, such as PDGFR, KIT and VEGFR are found to be activated in about 30% of pediatric HGG patients [81,82].

There are further reports about novel treatment strategies with even more limited patient numbers, which, therefore, did not qualify for this systematic review. Nimotuzumab, a monoclonal antibody, showed a favorable toxicity profile, even in prolonged use in pediatric HGG [83]. However, looking at the pediatric GBM subgroup of this study, prolonged survival time exceeding 40 months was not apparent, suggesting no benefit in survival time. The checkpoint inhibitor Nivolumab [84] predictably showed that the median survival for PD-L1 positive pediatric HGG patients was significantly higher than in PD-L1 negative patients. The response was, however, only transient and partial [84]. The lack of a significant effect of checkpoint inhibitors in pediatric GBM is also correlated to a known low mutational burden in pediatric GBM cells [85]. This leads to the assumption that a combination therapy with other immunomodulatory approaches, such as a combination of chimeric antigen receptor CAR-T cells and cancer vaccines could possibly better attack these therapeutically challenging tumors.

A patient group not included in the above-mentioned treatment options, besides the metronomic approach, are congenital and infant GBM patients. This subgroup of pediatric patients shows a different tumor biology and disease course with an extremely rapid growth and a highly vulnerable angiogenesis often leading to early and often fatal hemorrhage [15,16]. On the other hand, some cases show noticeable prolonged survival exceeding 24 months, treated by surgery and dose adjusted chemotherapy [28,86]. The vulnerable developing brain of infants under the age of three years leads to the fact that radiotherapy should be avoided, due to serious sequelae, such as developmental delay, endocrine dysfunction and secondary neoplasms of the CNS [15,16]. As the most common somatic alterations in infant HGG involve NTRK genes [17], larotrectinib, as a selective TRK inhibitor, represents a possible targeted therapy option here.

### 4.3. Novel Therapies within the Scope of Phase Ⅰ Trials and Future Perspectives

Further immunotherapeutic approaches under investigation include therapeutic vaccination, a treatment strategy to redirect T-cells against tumor antigens (Table 3). One category of a tumor vaccine is oncolytic viruses, such as herpes simplex virus (HSV) [87]. A phase I immunovirotherapy trial of oncolytic HSV-1 G207 with stereotactic placement of intratumoral catheters in pediatric HGG led to an increased number of tumor-infiltrating lymphocytes by bypassing the BBB without toxic effect [88]. Further oncolytic viruses under investigation are the parvovirus H-1 [89] and cytomegalovirus (CMV) [90]. After the detection of CMV in the majority of adult GBM cells [91,92], CMV antigens were also detected in approximately 66.7% of pediatric GBM samples [90]. CMV showed to enhance telomerase activity and angiogenesis in adult GBM, making it a possible target for immune-based therapy [93]. As autologous tumor lysate-pulsed dendritic cell vaccinations for pediatric patients with newly diagnosed or recurrent HGG showed feasibility and potential clinical benefit [94], these findings were combined leading to an ongoing trial (NCT03615404) investigating the feasibility of CMV RNA pulsed dendritic cells in children and adults. To our knowledge, to date, no results based on phase Ⅱ trials of these technologies in pediatric GBM patients exist. Another immunotherapeutic approach is CAR-T cells. A current phase I clinical trial is evaluating anti-IL13aR2 CAR-T cells in children 12 years and older with recurrent or relapsed gliomas (NCT0220836).

High-throughput genomic technologies discovered the biologic heterogeneity of several pediatric brain tumors [29,80]. This demands more individualized, targeted approaches [80]. Larotrectinib, a highly selective small-molecule inhibitor of tropomyosin receptor kinase (TRK) showed rapid and durable responses with a high tumor control rate and good tolerability in TRK fusion-positive primary central nervous system tumors in adult and pediatric patients [39]. In the pediatric cohort, the 24-week disease control rate was 69% (95% CI, 39−91%) for HGG. A pilot study investigating larotrectinib in newly diagnosed pediatric HGG with NTRK fusion is still recruiting, first results are expected in 2025 (NCT04655404). Kieran, et al. showed in a phase I trial the feasibility of targeting BRAF V600E mutated pediatric solid tumors, amongst others in pediatric GBM patients, with oral dabrafenib [95]. The effect of perifosine and temsirolimus in recurrent pediatric solid tumors by inhibition of the AKT and mTOR pathways axis, proved the feasibility and tolerable toxicity of the agent combination. However, in contrast to preclinical data, no objective response was observed [96]. Further phase I trials investigating the effect of INCB7839, an inhibitor of the ADAM 10 and 17 proteases (a disintegrin and metalloprotease) (NCT04295759) and ONC206, a selective dopamine-2 antagonist (NCT04732065) are ongoing. Evolving strategies to overcome the BBB, such as intraarterial delivery of BVZ and cetuximab are ongoing as well, showing good tolerance while Phase Ⅱ trials are needed to objectify the outcome [97].

Despite conducting a systematic review, several limitations are present in this study. First, we only searched two databases (Pubmed and Embase), while conference abstracts, review protocols, unpublished data and clinical evidence that was not indexed in bibliographic databases were excluded from this study. We further only searched for English literature, which carries a risk of omitting important data published elsewhere. Second, the included trials all comprised this highly heterogenous group of HGG consisting of anaplastic astrocytomas (WHO grade Ⅲ), diffuse midline gliomas and GBM (WHO grade IV). This pooling of patient cohorts is mainly explained through very limited patient numbers with limited observation time due to the aggressive biology of these tumors. However, within the review, we extrapolated the data for pediatric GBM patients wherever possible. Third, the primary outcome for the specific patient group of pediatric GBM varies in the included studies and is at times difficult to compare. Fourth, even within the GBM cohort the clinical and biological parameters, such as the extent of resection, the location of the tumor, previous treatment strategies and the molecular subgroups are highly heterogenous and at times data concerning these variables are not reported. Fifth, the patient cohorts within the studies are rather small, therefore conclusions need to be interpreted with caution. Finally, patients were recruited at different states of the disease and treatment, since the inclusion criteria within the studies were heterogenous, hampering the objective comparison of PFS or OS.

## 5. Conclusions

The role of pharmacological approaches for the treatment of pediatric GBM remains unsatisfying, since no novel approach led to improvement of PFS or OS. Novel treatment approaches, such as immunotherapeutic approaches and pharmacologically susceptible tumor-specific targets based on molecular biology, present promising preclinical results and should be the focus of future studies. Due to the limited number of patients, the molecular heterogeneity and aggressiveness of the disease, often leading to early death, multicenter, international trials are of paramount importance.

## Figures and Tables

**Figure 1 cancers-14-02814-f001:**
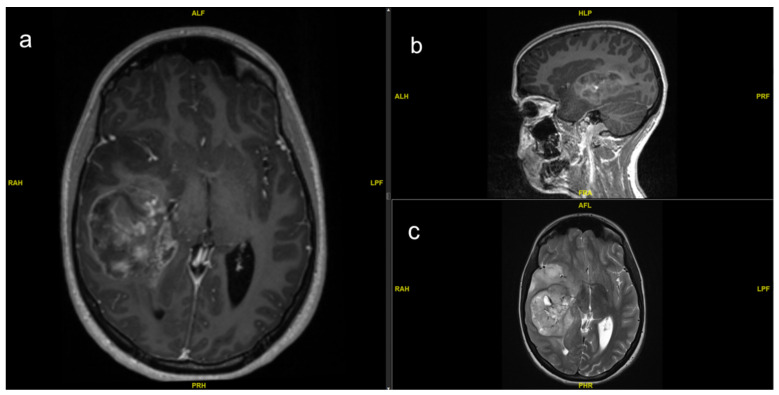
Preoperative MRI of a 14-year-old girl with right temporal GBM with heterogeneous enhancement in (**a**) axial and (**b**) sagittal T1-weight with contrast agent. The tumor mass causes a midline shift and shows central enhancement. (**c**) Axial T2-sequence with hyperintense signal of infiltrating tumor mass peripheral to enhancing center in the mesial and temporopolar structures.

**Figure 2 cancers-14-02814-f002:**
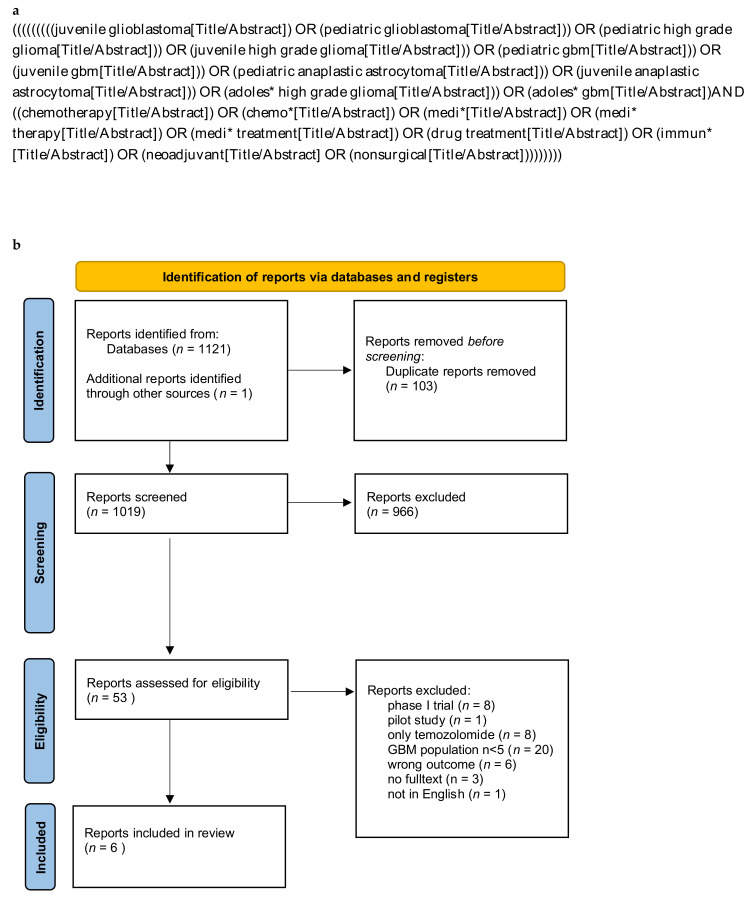
(**a**) Systematic search string used for Pubmed database. Adjustments of format have been made according to Embase guidelines. (**b**) PRISMA flow diagram (2020) for systematic reviews.

**Table 1 cancers-14-02814-t001:** Subdivision of pediatric GBM into three distinct molecular subgroups.

Category	Mutation/Cytogenetics	Age Distribution	Tumor Location	Prognosis
*H3*-Mutant	*H3*K27M *H3*G34R/V	younger children [42] adolescents and young adults [42,43]	almost exclusively in midline structures (=DMG) [42] cerebral hemispheres [29]	near 100% mortality [44] better OS than H3K27 mutations [44]
*IDH*-Mutant	*IDH* 1/2 mutation	older children/young adults [42,43]	cerebral hemispheres, frontal or temporal lobe [29]	better OS than *H3*-mutants [43]
*IDH/H3*-Wildtype	*BRAF*V600E, NF1 mutations, RTK fusions amplifications of EGFR, CDK6, MYCN PDGFRA and MET amplifications	infants/children/adolescents rare in children and adolescents [29] children and adolescents, rather rare [45]	supratentorial, commonly hemispheric [29,44] occur throughout the brain [44] hemispheric and midline [42]	increased survival [42,45] worst OS of WT group [42] poor OS [42,45]

H3 = histone 3, H3K27M = lysine-to-methionin mutation at position 27 in histone 3.1, 3.2 or 3.3; DMG = diffuse midline glioma, OS = overall survival, H3G34R/V = glycine-to-valin or arginine at position 34 in histone 3.3, IDH = isocitrate dehydrogenase, NF1 = neurofibromatosis type 1, RTK = receptor tyrosine kinase, EGFR = epidermal growth factor receptor, CDK6 = cyclin dependent kinase 6, MYCN = proto-oncogene, WT = wildtype, PDGFRA = platelet derived growth factor receptor alpha.

**Table 2 cancers-14-02814-t002:** Results of analyzed studies with objectifiable baseline characteristics.

Publication	Study Type	Recruitment Interval	pGBM Cohort (*n*)	Age at Study Entry ° (Years) (Median/Range)	Disease Status	Intervention	Primary Outcome GBM Cohort (EFS; PFS; OS)	Secondary Outcome (Toxicity; Toleration of Treatment)	Therapeutic Effect	Quality Assessment NOS/RoB-2 (points)
Gururangan et al., 2010 [51]	prospective phase-Ⅱ cohort trial	10/2006–09/2008	8	15.7 (5.6–20.1)	r/r	BEV plus irinotecan	2/8 SD at >12 weeks, of these 2 patients, median PFS: 8.3 months no sustained OR	20% toxicity with interruption of treatment	no efficacy in recurrent pGBM	5 (fair)
MacDonald et al., 2013 [50]	prospective phase-Ⅱ cohort trial	06/2008–12/2010	18	14.2 (1.1–20.3)	r/r	cilengitide	1/18 SD at 19 months	low toxicity rate, well tolerated	no efficacy in recurrent pGBM	5 (fair)
Robinson et al., 2014 [52]	multicenter, prospective phase-Ⅱ cohort trial	01/2005–03/2009	9	10 (0.6–21)	r/r	metronomic oral celecoxib, thalidomide, fenofibrate, low dose CPM and etoposide	1/9 SD at 27 weeks	low toxicity rate, well tolerated	no efficacy in recurrent pGBM	5 (fair)
Wetmore et al., 2016 [54]	multicenter, prospec-tive phase-Ⅱ cohort trial	01/2012–06/2013	7	14.5 (4.7–19.9)	r/r	sunitinib	Response rate (=CR or PR for at least 8 weeks): 0%	low toxicity rate, well tolerated	closing at interim analysis due to lack of efficiacy	6 (good)
Grill et al., HERBY trial 2018 [48]	randomized controlled trial	10/2011–02/2015	84	11 (3–17)	newly diagnosed	BEV	HR: 1.37 (95% CI 0.83 to 2.27) for RT plus TMZ compared to RT + TMZ + BEV	no safety concerns; more AEs in BEV-cohort	No measurable effect for unmethylated pGBM	some concerns (RoB-2)
Meng-Fen Su et al., 2020 [53]	multicenter, prospective phase-Ⅱ cohort trial	09/2009–08/2015	11	7.9 (3.2–19.9)	newly diagnosed	VPA and radiation followed by VPA and BEV	median EFS: 10.5 months median OS: 14.9 months	2 treatment interruptions after addition of BEV; RT and VPA with good tolerance	no improvement of OS	5 (fair)

° of the whole HGG cohort; pGBM = pediatric glioblastoma, NOS = Newcastle-Ottawa scale, RoB2 = Risk of Bias 2 tool, r/r = relapsed/refractory disease, PFS = progression free survival, EFS = event free survival, OS = overall survival, OR = objective response, CR = complete remission, PR = partial remission, HR = hazard ratio, CI = confidence interval, RT = radiotherapy, TMZ = temozolomide, BEV = bevacizumab, AEs = adverse events, VPA = valproic acid, CPM = cyclophosphamide.

**Table 3 cancers-14-02814-t003:** Currently recruiting trials and published phase I studies on new pharmacological treatment approaches.

Publication	Study Type	Conditions	Intervention	Outcome	Phase Ⅱ Proposed
Becher et al., 2017 [96]	multicenter	pediatric solid tumors	perifosine, temsirolimus	well tolerated, no objective response	NA
Kieran et al., 2019 [95]	multicenter	*BRAF* V600E mutation positive pediatric tumors	dabrafenib	well tolerated	yes
Friedman et al., 2021 [88]	multicenter	pediatric HGG	HSV-1 G207	well tolerated, objective change in tumor metabolism	yes
McCrea et al., 2021 [97]	multicenter	HGG and DIPG	intraarterial BEV and cetuximab with BBB disruption	well tolerated, little objective effect	yes
NCT04295759	multicenter	pediatric HGG	INCB7839	recruiting	NA
NCT04732065	multicenter	pediatric brain tumors	ONC206	recruiting	NA
NCT04655404	multicenter	pediatric HGG	larotrectinib	recruiting	NA
NCT03615404	single center	pediatric brain tumors	CMV-DC with GM-CSF	completed, publication pending	NA
NCT02208362	single center	pediatric and adult glioma	IL13Ralpha2-CAR-T cells	active, not recruiting	NA

HGG = high grade glioma; DIPG = diffuse intrinsic pontine glioma; *BRAF* V600E = v-Raf murine sarcoma viral oncogene homolog B1 V600E; BEV = bevacizumab; BBB = blood–brain barrier; INCB7839 = aderbasib; CMV-DC = cytomegalyvirus infected dentritic cells; GM-CSF = granulocyte macrophage colony stimulating factor.

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
