# Peer review of "Novel Pharmacological Treatment Options in Pediatric Glioblastoma—A Systematic Review"

_cancers, 2022, doi:10.3390/cancers14112814_

Round 1

Reviewer 1 Report

Wyss et al. designed a systematic review to explore the current interventions and their effectiveness/harm in the treatment of paediatric glioblastoma. Primary outcomes and toxicities were evaluated as outcomes. Overall, the manuscript is well written with a clear description of methods and results. The study can be improved further with a few details listed below:

1. In the introduction, please include a few sentences for the most common signalling pathways that are being targeted in the treatment of paediatric GBM (in addition to their last paragraph talking about molecular subtypes).

2. Please make a chart/table etc for a clear description of PICO.

3. Were the authors able to extract the secondary outcomes from the studies they included in their systematic review?

4. Please clarify why January 2010 was selected as the start date (lack of substantial studies etc)

5. Authors mentioned in the Discussion that they only used PubMed and Embase in their literature search. Please also mention that conference abstracts, review protocols, unpublished data, and clinical evidence that was not indexed in bibliographic databases were excluded from this study.

6. Minor recommendation: In Fig. 1b, the authors could include “Eligibility” category in between the Screening and Included sections.

Reviewer 2 Report

To editors and reviewers
Novel pharmacological treatment options in pediatric glioblastoma - a systematic review
- This is an very interesting manuscript that can be considered for publication in CANCERS. The manuscript is appropriate with aims and scope of journal.
- I suggested some revisions below and after revisions the manuscript can be published.
1) Some citation and references are not precise as MDPI format. Please check and revise.

2) This is a review. Unstructured abstract will be appropriate.

3) In children, infratentorial tumors are more common than supratentorial tumors (PMID: 31666838). Please also write an imaging paragraph in introduction part to diagnose pediatric brain tumor (PMID: 32366451)

4) Limitation part is not an independent part of a review paper. Delete subheading.

5) Add MRI images to illustrate pediatric GBM.

Sincerely

Reviewer 3 Report

The authors wrote a systematic review based on a limited number of clinical trials that determined the effectiveness of new drugs to treat pediatric GBM and HGG. This review of the literature is relevant since there is no effective treatment. They conclude that none of the new treatment modalities demonstrated a positive effect on overall survival or progression-free survival.

The analyzes were carried out according to the rules of the art, while the discussion provided relevant comments that could explain the disappointing results obtained in pediatric patients. New treatments, such as immunotherapy, have been summarized and merit future study.

Minor comments

Q1: Define the abbreviation HGG.

Q2: Define the abbreviation “TRK » the first time it appears.

Q3: Define the abbreviation VPA only once in the main text.
